# Microstructural Assessment, Mechanical and Corrosion Properties of a Mg-Sr Alloy Processed by Combined Severe Plastic Deformation

**DOI:** 10.3390/ma16062279

**Published:** 2023-03-12

**Authors:** Ruslan K. Nafikov, Olga B. Kulyasova, Ganjina D. Khudododova, Nariman A. Enikeev

**Affiliations:** 1Institute of Physics of Advanced Materials, Ufa University of Science and Technology, 32 Zaki Validi Str., 450076 Ufa, Russia; 2Laboratory of Metals and Alloys Under Extreme Impacts, Ufa University of Science and Technology, 32 Zaki Validi Str., 450076 Ufa, Russia; 3Laboratory of Multifunctional Materials, Ufa University of Science and Technology, 32 Zaki Validi Str., 450076 Ufa, Russia; 4Laboratory for Dynamics and Extreme Performance of Promising Nanostructured Materials, Saint Petersburg State University, 199034 St. Petersburg, Russia

**Keywords:** biodegradable magnesium alloy, ultrafine-grained materials, severe plastic deformation, microstructure, mechanical properties, corrosion

## Abstract

The development of high-performance biodegradable alloys with controllable corrosion rates to be used for manufacturing advanced implants is a hot topic of modern materials science and biomedicine. This work features the changes in microstructure, corrosion behavior and mechanical properties of the Mg-2 wt.%Sr alloy progressively induced by equal-channel angular pressing, high-pressure torsion and annealing. We show that such processing leads to significant microstructure refinement including diminishing grain size, defect accumulation and fragmentation of the initial eutectics. We demonstrate that the application of severe plastic deformation and heat treatment is capable of considerably enhancing the mechanical and corrosion performance of a biodegradable alloy of the Mg-Sr system. The best trade-off between strength, plasticity and the corrosion resistance has been achieved by annealing of the Mg-Sr alloy subjected to combined severe plastic deformation processing.

## 1. Introduction

Biodegradable metals and alloys have attracted a great deal of attention from modern researchers in materials science and biomedicine thanks to their highly attractive biomedical and mechanical properties, having substantial advantages over the bioinert materials traditionally used for manufacturing orthopedic implants [1]. Among the biodegradable materials, magnesium (Mg) and its alloys have become a very popular topic in the field, an yielding ever-growing number of publications as summarized in [1,2] and in the latest overviews observing the most recent findings [3,4,5,6,7]. Mg-based materials are mechanically very compatible with human tissues; the Young’s modulus of Mg (*E* = 41–45 GPa) and the density (*ρ* = 1.74–1.84 g/cm^3^) are similar to those of bone (*E* = 15–25 GPa and *ρ* = 1.8–2.1 g/cm^3^), and using Mg alloys can reduce the undesired stress shielding effect [8]. Mg-based alloys are the most lightweight among biomaterials. Furthermore, Mg is a vitally important chemical microelement required for metabolism; it has good biocompatibilty, biodegradability and absorbability, and it demonstrates high strength in comparison to polymers and high ductility in comparison to ceramics [2,8], which promotes its application as a material for bioresorbable implants [3,4,5,6,8].

Numerous alloying systems (with additions of Al, Mn, Zn, Ca, Li, Zr, Y, rare earth elements and so on) have been developed to the enhance mechanical and functional performance of Mg with great focus on controlling its corrosion rate, which is one of the most important issues from the viewpoint of its potential biomedical application [1,2,3,4,5,6,7]. Recently, another Mg system alloyed with Sr was proposed [9]. Sr can improve the mechanical properties of Mg, facilitate grain refinement processes and increase the corrosion resistance. Alongside this, Sr has unique pharmacological effects and can be used in the treatment of osteoporosis as it draws extra Ca to the bone and stimulates its regeneration [10,11]. The alloy exhibits excellent results in in vitro and in vivo studies [9], and Mg appears to be an ideal platform for the Sr delivery into human tissues [12].

Additional routes to enhance the functional properties of metallic alloys can be related to the microstructural design. Nanostructuring by severe plastic deformation (SPD) is believed to be a powerful approach to regulate fine structures of bulk metallic materials to significantly improve their mechanical and functional properties [13]. SPD was found to be very fruitful in enhancing the properties of Mg-based alloys [14]. The application of this concept for the Mg-Sr system demonstrated, for the first time, its high efficiency to form homogeneous UFG structures by SPD via high pressure torsion (HPT) [15,16] and equal-channel angular pressing (ECAP) [17], allowing significant impacts on mechanical strength and a lowered degradation rate of the alloy (unlike the case of pure Mg). SPD Mg-Sr alloys also proved to have high biocompatibility, including new bone formation, examined by in vitro and in vivo studies [16,17]. However, producing the UFG structures that provide the optimal mechanical and functional (first of all, corrosion) performance of Mg-Sr alloy is still a challenging task for SPD researchers since it can be defined not only by the grain refinement of the Mg matrix but also by structural homogeneity, defect configuration, morphology and distribution of the Mg_17_Sr_2_ eutectic and so on [17,18]. For example, HPT provides very high strengthening with yield stress raised up to 166 MPa of a Mg-2Sr alloy [15,16], but leads to almost brittle behavior with elongation to failure not exceeding 1.5%. ECAP allowed increased elongation to failure by two times compared with the coarse-grained state of a Mg-Sr alloy (up to 8%); however, the achieved yield stress was found to be not higher than 110 MPa [17].

Combinations of different SPD techniques and heat treatments can provide additional capabilities to vary the produced UFG states [13]. In particular, the combined ECAP + HPT route was found to be more effective in microstructure refinement than HPT alone [19]. Application of ECAP could also change the eutectic morphology prior to HPT; this effect was shown to be important in processing immiscible Al alloys [20]. In the present study, we report on the effect of combined ECAP + HPT processing followed by annealing in order to achieve different UFG structures in a Mg-Sr alloy. As shown in [9], introducing different concentrations of Sr into Mg progressively increased the mechanical performance of the alloy as well as its corrosion resistance, with the best combination of properties with a Sr content of 2 wt.%. Exceeding this value, the alloy demonstrated degradation of the target properties. Therefore, the Mg-2 wt.%Sr alloy was chosen as the subject of this study.

## 2. Materials and Methods

A binary Mg alloy containing 2 wt.% of Sr was selected as the object of investigation, which hereinafter will be referred as a Mg-2Sr alloy. The as-received cast workpieces of the Mg-2Sr alloy had been provided in the form of cylinders with a diameter of 20 mm and a length of 100 mm by Professor Yufeng Zheng from the School of Materials Science and Engineering, Peking University (China). These workpieces were homogenized by annealing in a muffle furnace Nabertherm N321/13 (Nabertherm GmbH, Lilienthal, Germany) at T = 450 °C for 24 h followed by cooling into water.

The microstructure refinement of the as-received material was achieved through two popular SPD techniques: ECAP and HPT. ECAP was performed using a die-set with a channel intersecting angle of 120°; the processing temperature was 400 °C. Workpieces were subjected to four ECAP passes via route Bc, which implies the rotation of a workpiece by 90° around its axis between the passes (Figure 1a). This ECAP route was shown to be the most effective in microstructure refinement due to the deformation path change [21]. Furthermore, the discs with a diameter of 20 mm and a thickness of 1.8 mm were cut out from the ECAP-processed workpieces and subjected to HPT at a temperature of 100 °C under an applied pressure of 6 GPa. The number of HPT revolutions was five, with the anvil rotation rate of 1 rev/min. This state processed by combined SPD techniques is denoted below as the “cSPD” state. Then, the HPT-processed samples were annealed at a temperature of 250 °C for 1 h followed by cooling in air. This state is referred to below as the “cSPD AA” state.

For microstructural studies, the specimens were ground with sandpapers with progressively changing abrasivity from 600 to 1200 grit, followed by final polishing with a water solution of a chromium (III) oxide. The sample surface was etched in a solution containing 2.5 mg of picric acid, 2.5 mL of acetic acid, 5 mL of distilled water and 50 mL of ethanol during a period of 10 s with subsequent washing in ethanol.

The specimens’ microstructure was characterized with the help of the JEM-6390 (JEOL Ltd., Tokyo, Japan) scanning electron microscope (SEM) and the JEM-2100 (JEOL Ltd., Tokyo, Japan) transmission electron microscope (TEM) with accelerating voltages of 10 kV and 200 kV, respectively. The macrostructure was observed with the optical microscope Olympus GX51 (Olympus Corporation, Tokyo, Japan). The average values of the grain size were calculated through the grain intercept method counting at least 300 grains per each state.

Fine microstructure observations were carried out using specimens prepared by jet electropolishing with a TenuPol-5 device (Struers LLC, Cleveland, OH, USA) in the electrolyte containing 30% of nitric acid and 70% of methanol. The thin foils for the TEM studies were cut out from the area corresponding to the middle of the disc or the cylinder radius for all the investigated states. For the HPT state, the specimens were cut out in the normal direction to the disc surface, while for the ECAP state, the specimens were prepared from the cross-section normal to the extrusion direction.

X-ray diffraction (XRD) analysis was conducted using the D2 Phaser benchtop X-ray diffractometer (Bruker AXS GmbH, Karlsruhe, Germany) with the CuKα radiation at 30 kV and 10 mA, a scan step of 0.02° and a scanning rate of 1°/min. The XRD analysis of the recorded X-ray diffraction patterns was performed using the Rietveld refinement method implemented in the MAUD software package [22]. Considering the crystallographic features of the Mg-2Sr alloy, the anisotropic Popa line broadening model was applied to account for the peculiarities of HCP symmetry, which provides different changes in peak profiles belonging to different families of crystal planes depending on the density of dislocations with the different values of the Burgers vector [23]. As a result, the values of coherent domain size (*d*) and crystal lattice microstrain (<*ε*^2^>^1/2^) were calculated for the peaks from the corresponding reflection planes.

The deformation of HCP crystals leads to the generation of <*a*> and <c + a> dislocations with Burgers vectors of 1/3<11$\overset{-}{2}$0> and (1/3<11$\overset{-}{2}$3>, respectively. Dislocations of the *c*-type (Burgers vector [0001]) may arise from reactions between the dislocations above. Resolving these dislocations to their *c*- and *a*- components, one can deduce their density from the integral broadening of reflections corresponding to the basal and prismatic planes [24].

Further, the densities of dislocations with the Burgers vector values *b_a_* and *b_c_* were estimated via the following expressions [24]:(1)$$\rho_{a}=\frac{K_{a}\varepsilon_{a}^{2}}{b_{a}^{2}}/\ln\frac{d_{a}}{2r_{0}}$$
and
(2)$$\rho_{c}=\frac{K_{c}\varepsilon_{c}^{2}}{b_{c}^{2}}/\ln\frac{d_{c}}{2r_{0}},$$
where *K_a_* and *K_c_* are constants (*K_a_* = 52.1, *K_c_* = 26.1) and *r*_0_ = 1 nm [24], *b_a_* and *b_c_*, *d_a_* and *d_c_*, *ε*_a_ and *ε_c_* are values of the Burgers vectors, coherent domain size and root mean square strain for the *a*- and *c*-components, respectively. The value for *b_a_* was taken as 1/3<11.0> and for *b_c_* as [0001]. Coherent domain size and microstrain values were calculated by refining peak profiles for families of the crystallographic planes (10.0)/(11.0) and (00.2), respectively. First estimations showed that *ρ_a_* >> *ρ_c_* (by at least one order of magnitude), which is consistent with the XRD analysis of other HCP materials subjected to SPD, such as Ti [25], where it was shown that the main deformation activity is associated with the basal slip. Therefore, in the following considerations, we analyzed the SPD-induced changes in the dislocation structure based on *ρ_a_* calculation.

Microhardness testing was carried out using a microhardness tester Omnimet (Buehler Ltd., Lake Bluff, IL, USA) in accordance with the state standard GOST 2999-75. We measured the Vickers hardness along the sample diameter with an increment of 1 mm, a loading time of 10 s and a load of 0.050 kgf.

Due to the limited dimensions of SPD-produced workpieces, tensile tests were carried out using miniaturized specimens with the design presented in Figure 1 with the help of a universal testing machine Instron 5982 (ToolWorks Inc., Norwood, MA, USA). The sample gauge was *l*_0_ = 4 mm long, the width was 0.76 to 0.96 mm and the thickness was 0.83 to 0.96 mm. Similar specimens have been specially designed to evaluate the mechanical properties of SPD materials [26] and are widely used in many related studies. It is accepted that strength parameters are captured well with the help of these specimens as compared with standard test results; however, the total elongation to failure values can be considerably affected by the sample size effect [27,28]. It must be emphasized that the reported values can be correctly interpreted in terms of relative comparison between the investigated alloy’s states in frames of this study and should be compared with the literature data with caution. To ensure the reproducibility of measurements, at least 3 specimens were tensile-tested for each structural state.

The corrosion behavior of the Mg-2Sr alloy was investigated according to the ASTM-G1-03-E standard. Before the corrosion tests, the specimens were ground with sandpapers and finally polished with a water solution of chromium (III) oxide as described above, to negate the effect of surface roughness. Mass loss values were measured after immersion of the samples into Ringer’s solution (composition: 8.6 g/L NaCl; 0.3 g/L KCl; 0.25 g/L CaCl_2_, pH 7) at a temperature of 37 °C. Every 24 h after immersion, the samples were removed, washed and weighed using an Explorer analytical balance (Ohaus Corporation, Parsippany, NJ, USA) with an accuracy of 0.1 mg. Prior to mass measurements, corrosion products (Mg(OH)_2_) were removed from the surface by washing in an ultrasonic bath in a solution of the following composition: 200 g of CrO_3_, 10 g of AgNO_3_, 20 g of Ba(NO_3_)_2_ and 1000 mL of H_2_O. Ringer’s solution was changed every week. For the sake of reproducibility of measurements, 2 samples were tested for each state.

The corrosion rate *CR* (mm/y) was calculated in accordance with the ASTM G3–63592 standard as:(3)$$CR=\frac{87.6(M_{0}-M_{1})}{St\rho },$$
where *CR* is the corrosion rate (mm/y); *S* is the sample surface area, cm^2^; *M*_0_ is the initial mass (mg), *M*_1_ is the mass after immersion (mg); *t* is the exposure time, h; *ρ* is the metal density, g/cm^2^. The surface area was estimated as *S* = 2π*r*^2^ (where *r* is the specimen’s radius). The total immersion time consisted of 30 days.

## 3. Results

### 3.1. The Effect of ECAP, cSPD and Annealing on the Microstrcuture of Mg-2Sr Alloy

The microstructure of the Mg-2Sr alloy in the initial state (Figure 2) comprised a solid solution (α-Mg) and an eutectic (α-Mg + Mg_17_Sr_2_) forming network structures as also pointed out in [29]. Similar to the data reported by [9], it had a coarse-grained structure with a mean grain size of 227 ± 21 μm and low dislocation density.

ECAP processing resulted in significant microstructure refinement of the Mg-2Sr alloy (Figure 2b). This image was achieved in backscattered electron mode, while all others were collected in secondary electrons. The mean grain size was reduced to 10 ± 2 μm, while the eutectic was considerably fragmented into separate elongated pieces. Further dramatic grain refinement was achieved following HPT at 100 °C (the cSPD state)—the grain structure could not be resolved due to the limits of the SEM resolution (Figure 2c). The eutectic had been notably fragmented, forming coagulated globular structures. The spherical form of the intermetallic fragments allowed the estimation of their volume fraction, which comprised 12 ± 1%. Further annealing at 250 °C led to considerable recrystallization—the SEM-defined grain size increased up to 6.7 ± 1.7 μm (Figure 2d), while the eutectic retained the globular morphology and its volume fraction increased up to 14 ± 2%.

These data are consistent with the results of TEM studies (Figure 3) performed for ECAP and cSPD states; in the latter state, SEM could not reveal fine features of grain and defect structures. Figure 3a indicates that ECAP entailed the formation of developed dislocation configurations with spacings of about a few hundred nanometers arranged in a kind of cell structure inside the large grains detectable using SEM (Figure 2b) as well as large dislocation agglomerations in the form of a dislocation forest (Figure 3b). Figure 3c proves that cSPD leads to much more pronounced grain refinement—a homogeneous ultrafine-grained structure was formed with a mean grain size of 0.25 ± 0.05 μm as determined using TEM, while the dark field image testified that the fragments represent grains with high angle grain boundaries (Figure 3d).

### 3.2. X-Ray Analysis of the Effect of ECAP, cSPD and Annealing on Microstrcuture of the Mg-Sr Alloy

The X-ray profiles for the Mg-2Sr alloy in different structural states are presented in Figure 4. Qualitative phase analysis indicated that all states consist of the α-Mg phase and the Mg_17_Sr_2_ s phase. The correct quantitative phase analysis was complicated by deformation texture effects for the states produced via different deformation modes (such as ECAP and HPT), while the coarse-grained state might have the statistically limited number of grains oriented in the reflection position with respect to the scattering vector.

X-ray analysis data presented in Table 1 confirm that both ECAP and cSPD induce high dislocation density in the Mg-2Sr alloy, significantly decreasing coherent domain size (*d_xrd_*). Annealing led to considerable recovery of microstructure: the dislocation density decreased by an order of magnitude. We also observed changes in the lattice parameters of the Mg matrix (*a*, *c*) after ECAP and cSPD, which might be associated with strain-induced dissolution of the alloying elements belonging to the particles. Annealing recovered values of the lattice parameters indicating possible precipitation processes. These considerations are consistent with the observations on the second phase evolution in Section 3.1. Such phenomena caused by enhanced atomic transport are often reported in SPD-related literature, and their analysis requires separate dedicated studies.

### 3.3. Hardness and Mechanical Properties

The results of the assessment of the hardness and mechanical properties of the Mg-2Sr alloy in different structural states are summarized in Table 2. ECAP and cSPD of the coarse-grained Mg-2Sr alloy resulted in the considerable increase in microhardness—from 38 HV to 55 and 65 HV, respectively. This correlates well with the microstructural changes—progressing grain refinement and boosting the dislocation density, which are the most pronounced in the cSPD state. Annealing after cSPD reduced the microhardness of the cSPD-AA state down to 52 HV due to recrystallization; however, it kept the value corresponding to the ECAP state within inaccuracy of measurements.

Deformation tensile curves presented in Figure 5, and the mechanical parameters shown in Table 2 are in good agreement with the microhardness measurements. ECAP of the initial Mg-2Sr alloy provided more than twofold increase in yield stress and threefold increase in ultimate tensile strength (UTS). HPT additionally impacted these values by approximately 50%, reaching the highest UTS values of 253 MPa. It is important to note that ECAP considerably enhanced the plasticity of the initial material, while the cSPD state demonstrated almost brittle behavior. The annealing of cSPD state did not notably change the UTS value and alongside this, it dramatically improved elongation to failure, which reached the maximum registered level (6.5%) among all considered cases.

### 3.4. Corrosion Behavior

Images of the specimens prior to the corrosion tests and after progressive exposure in Ringer’s solution (for 1, 7, 16 and 30 days) are shown in Figure 6, and the quantitative data on the corrosion rate are summarized in Table 3 and Figure 7. Figure 6 visibly illustrates that the initial Mg-2Sr alloy demonstrated the highest corrosion rate among all the studied states—the first corrosion-induced pits appeared 24 h after immersion. The corrosion rate showed a tendency to decrease with immersion time (Figure 7) with a saturation value of about 0.91 mm/year.

The ECAP state did not exhibit any clearly noticeable defects after immersion for less than 9 days. Afterwards, local pits formed in the center of specimens and subsequently propagated to the periphery; the saturation corrosion rate was 0.24 mm/year, which is three times less than the initial state. Compared to the ECAP state, the Mg-2Sr subjected to cSPD showed a somewhat higher corrosion rate within the early stage of immersion (Figure 7), with corrosion pitting appearing after an exposure time of 10 days (Figure 6). However, the corrosion rate then saturated to the same level as the ECAP state (0.27 mm/year).

The annealing of the cSPD state reduced its corrosion resistance, characterized by the earlier onset of corrosion pitting (after immersion time of 7 days) with a saturation corrosion rate of 0.44 mm/year.

Decreasing the corrosion rate of the Mg-2Sr in all studied structural states (Figure 7) with the immersion time can be related to developing passivation and formation of the Mg(OH)_2_ corrosion product on the specimens’ surface which is consistent with the data reported in [25].

SEM studies of the initial specimens’ surface after an immersion time of 30 days (Figure 8a) revealed pronounced intergranular corrosion with large damages in the area of coagulated eutectic fragments. In the cSPD-AA state, the corrosion damages are also related to the eutectic; however, the eutectic fragments are notably smaller and more uniformly distributed across the alloy’s surface, diminishing the corrosion damages and providing more uniform pitting corrosion.

The ECAP state of the Mg-Sr alloy is characterized by high densities of intragranular dislocations, producing elastic distortions of the Mg crystal lattice. These internal stresses assist forming passivation films that block corrosion.

The positive effect of grain refinement on the corrosion resistance of Mg alloys has already been reported in the literature [30,31,32] and related to more uniform distribution of surface defects such as dislocations and grain boundaries. In the case of the Mg-Sr system, an additional factor to increase the corrosion resistance in the deformed states can be attributed to the fragmentation and partial dissolution of the second phases due to SPD, which is consistent with the findings reported in [18]. 

## 4. Discussion

We have studied the effect of the combined processing by ECAP and HPT as well as subsequent heat treatment on the microstructure, mechanical and corrosion properties of the Mg-2Sr alloy. The collected experimental data showed that both ECAP and cSPD led to progressive grain refinement and introducing high dislocation densities, which provided a significant increase in hardness and strength. ECAP induced dislocation agglomerations consisting of pile-ups and a dislocation forest visible inside large grains. The application of cSPD qualitatively transformed the ECAP-formed microstructure into the UFG state with a highly refined grain size (down to 250 nm). These processes were accompanied by the significant fragmentation and re-distribution of the Mg_17_Sr_2_ eutectic with a transition to formation of globular dispersed second phase fragments after cSPD. As a result, cSPD provided maximal values of tensile strength and microhardness, while ductility was almost negligible, while ECAP led to a considerable increase in elongation to failure as compared with the initial state. The enhancement of plasticity of the Mg-2Sr alloy by ECAP was also observed earlier [17] and it can be explained by the more homogeneous distribution of the refined second phase as well as by grain refinement, which led to the formation of new grains that could be oriented favorably with respect to the loading condition. This can provide better accommodation of deformation, which can be troublesome in the coarse-grained state due to the limited number of slip systems in Mg, especially when it is tensile-tested using miniature specimens. A radical reduction in plasticity of the Mg-2Sr alloy after cSPD can be explained by high internal stresses (consistent with XRD analysis, Table 1) in the UFG state with a low number of dislocations resolvable in the grain interiors as shown through TEM (Figure 3b,c). The plastic deformation of this state leads to early fracture due to critical stresses accumulated in the interfaces. This observation is also in agreement with the studies of Mg-Sr alloys subjected to HPT.

The annealing of the cSPD state resulted in significant structure relaxation expressed both by the internal stresses release and an increase in coherent domain/grain size. This dramatically improved the elongation to failure of the Mg-2Sr alloy as compared with the cSPD state. Assuming that the tensile strength was insignificantly reduced, this state represented the best trade-off in terms of strength and plasticity among all the studied states. This effect might be due to the relaxation of the refined grain structure reinforced with precipitates and fragments of the second phase.

Note that plain HPT [15] did not allow us to achieve strength values in the Mg-2Sr alloy as high as in the cSPD and, moreover, the annealing of HPT-processed state even at 200C considerably diminished its strength [15], while the cSPD-AA processed Mg-2Sr alloy preserved its UTS at a level comparable with the cSPD state (Table 2). We relate this difference to the effect of ECAP prior to HPT; it allowed significant fragmentation of the second phase eutectic and facilitation of the following HPT-induced structural transformation. The revealed important effect of the morphology of the intermetallic phase on microstructure evolving during subsequent SPD is consistent with the results reported for immiscible Al alloys [20,33].

The observed change in lattice parameters in the differently treated Mg-2Sr alloy can be related to the evolving residual stresses and/or changing of the Sr content in Mg. We suggest that the observed changes in the lattice parameters can be associated rather with the effects of solid solution transformation. This is consistent with the SEM data on the changes in the fraction of the secondary phase after cSPD and annealing (see Figure 2c,d). Sr has up to 0.11 wt.% solubility in Mg [4] and severe deformation could promote mechanical mixing in the process of the eutectic fragmentation assisted with diffusion enhanced under SPD conditions. The solid solution decomposition can occur through precipitation as well as via segregation and aggregation of solutes as reported for Mg alloys in the literature [34], but this is not traceable using TEM techniques. Presumably, there could be possible effects of the lattice parameter change related to deformation-induced texture in terms of preferable oriented specific defect configurations. This issue requires future careful investigation.

The observed saturation in the corrosion rate (Figure 6) might occur because of the development of passivation processes and formation of the Mg (OH)_2_ corrosion product on the specimens’ surface. Careful examination of the corrosion product effect is required to discuss these effects in detail, since the corrosion rate can depend on numerous structural features and their interaction during the immersion. For the purpose of the present manuscript, we tried to highlight the principal difference in the corrosion behavior among the different structural states of the alloy revealed under the same testing conditions. Systematic studies involving polarization and impedance tests are planned to be reported in a separate dedicated study.

The results of the immersion tests confirmed that the galvanic effect is promoted in the interface area near Mg_17_Sr_2_ fragments [18], which facilitates the corrosion of the Mg matrix adjacent to the intermetallic phase. With immersion time, corrosion-induced degradation gradually spreads from to the interior of Mg grains, which is consistent with the results reported in [35]. The highest corrosion rate of 0.91 mm/y after 30 days of immersion was observed in the coarse-grained state with manifestation of large and deep corrosion pits on the surface after a week of immersion. In contrast, the alloy after ECAP exhibited a minimal saturation corrosion rate of 0.24 mm/y. Figure 6 shows that the samples after ECAP have a totally different morphology; root-like corrosion grooves can be observed on the surface. Mg-2Sr alloy in the cSPD state demonstrates localized shallow corrosion pits, as well as small areas of the matrix experiencing traces of degradation.

The annealed cSPD state undergoes corrosion occurring in the regions of the coagulated eutectic (Figure 8d). Pitting corrosion, being uniform across the whole sample surface, forms near the second-phase fragments. With immersion time, it evolves into the percolating pitting corrosion, since the eutectic’s potential is less than the potential of the Mg matrix in these areas. The corrosion rate of the cSPD-AA state exceeds the value for the cSPD state, which is consistent with the higher fraction of intermetallic phase fragments revealed for the former case.

Comparing the combinations of properties exhibited by every investigated case, it can be summarized that the cSPD-AA state possesses the best combination of strength, plasticity and corrosion resistance. The presented findings show a pathway for the microstructural design of high-performance biodegradable alloys with controllable corrosion rate as an addition to commonly accepted strategies [36]. However, further investigations to account for the effects of precipitation, segregation and strain-induced texture as well as versatile corrosion studies are required to explore the findings of the present study.

## 5. Conclusions

A Mg-Sr biodegradable alloy was progressively subjected to different deformation and heat treatments including ECAP, HPT and annealing. The produced states of the alloy exhibited markedly different microstructures, which provided notable changes in their mechanical and corrosion behavior.

The combined SPD with ECAP followed by HPT led to extreme grain refinement with formation of the UFG structure with a grain size of 250 nm and fragmentation of eutectic into the globular homogeneously distributed particles. This impacted the tensile strength of the alloy (to 253 MPa) accompanied with the abrupt reduction in plasticity. The annealing of this state at 250 °C resulted in the stress relaxation and precipitation of intermetallic particles. The tensile strength remained at the level of 250 MPa and elongation to failure reached 6.5%, which comprised the highest value among all the studied cases.

Immersion tests showed that all the SPD-processed states had better corrosion resistance than the coarse-grained alloy, which may be explained by the formation of more homogeneous microstructure and the fragmentation of eutectics.

The best combination of strength, plasticity and corrosion resistance was achieved in the alloy subjected to the combined SPD and post-deformation annealing. Thus, the SPD-driven microstructural design proved to be a powerful technique to control the mechanical and corrosion performance of a Mg-Sr biodegradable alloy.

## Figures and Tables

**Figure 1 materials-16-02279-f001:**
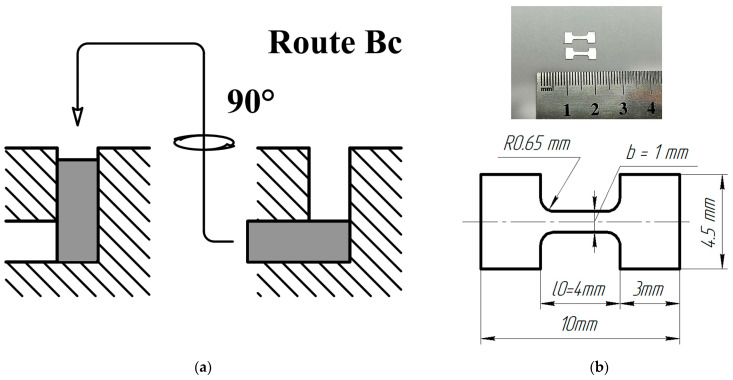
Schematic representation of ECAP process according to route Bc (**a**); miniature tensile specimen design: a general view at the top and specimen layout at the bottom (**b**).

**Figure 2 materials-16-02279-f002:**
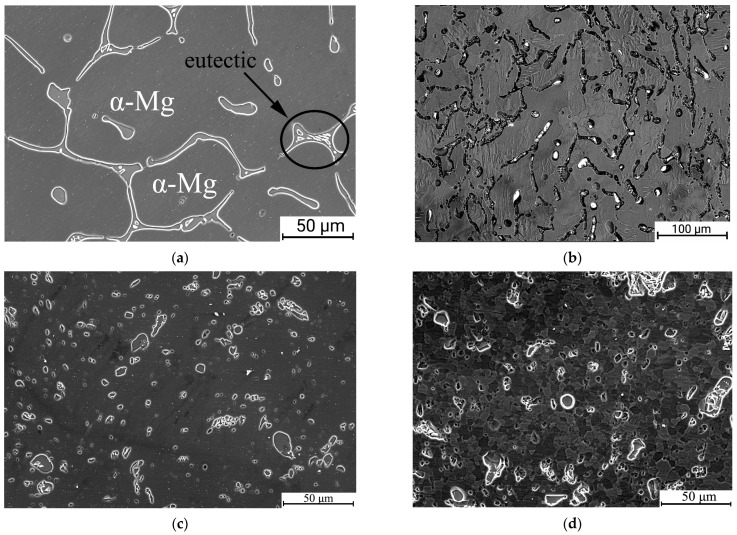
SEM images revealing microstructure of the Mg-2Sr alloy in the different states: initial (**a**); ECAP (**b**); cSPD (**c**) and cSPD-AA (**d**).

**Figure 3 materials-16-02279-f003:**
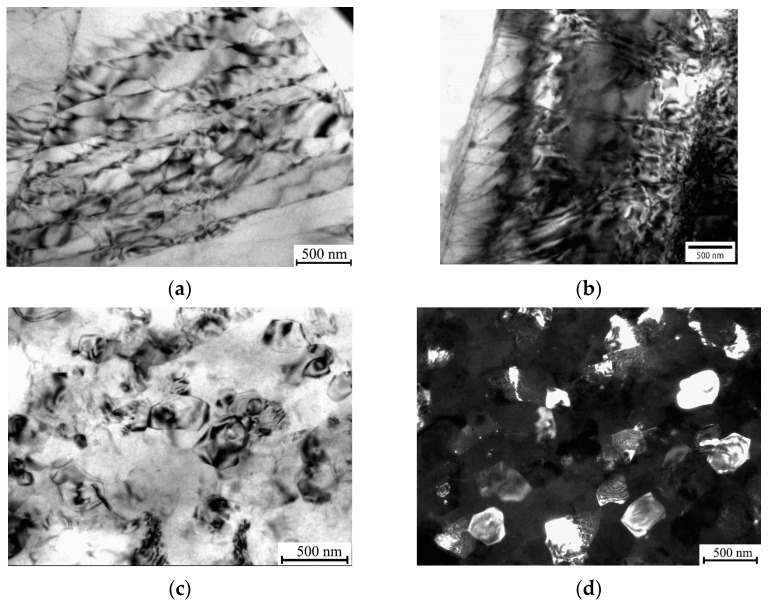
TEM images revealing microstructure of the deformed Mg-2Sr alloy: after ECAP, bright field images showing dislocation networks (**a**) and forest (**b**); after cSPD, bright field (**c**) and dark field (**d**) images.

**Figure 4 materials-16-02279-f004:**
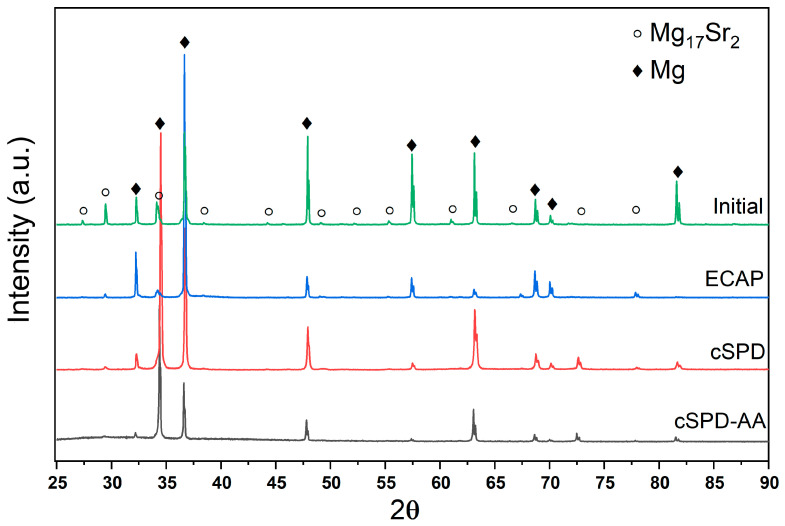
X-ray diffraction patterns of the Mg-2Sr alloy in different structural states.

**Figure 5 materials-16-02279-f005:**
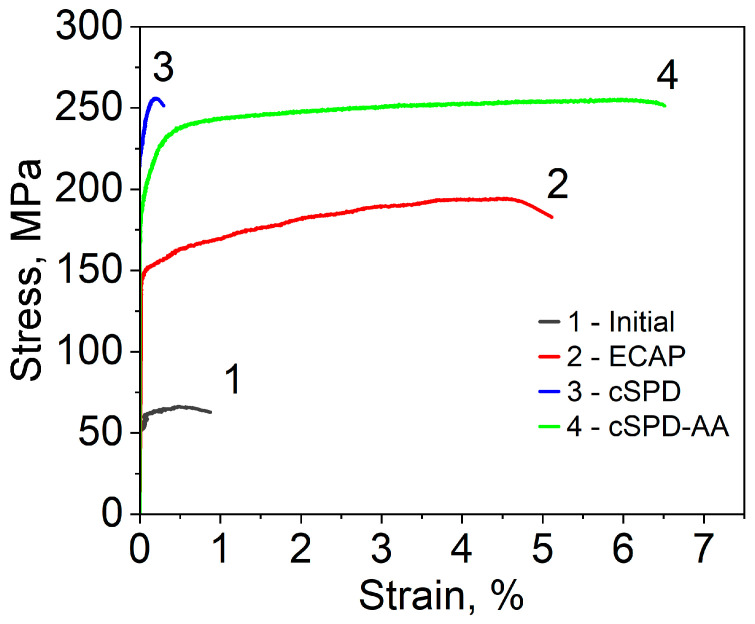
Engineering “stress–strain” curves for the Mg-2Sr alloy in different structural states.

**Figure 6 materials-16-02279-f006:**
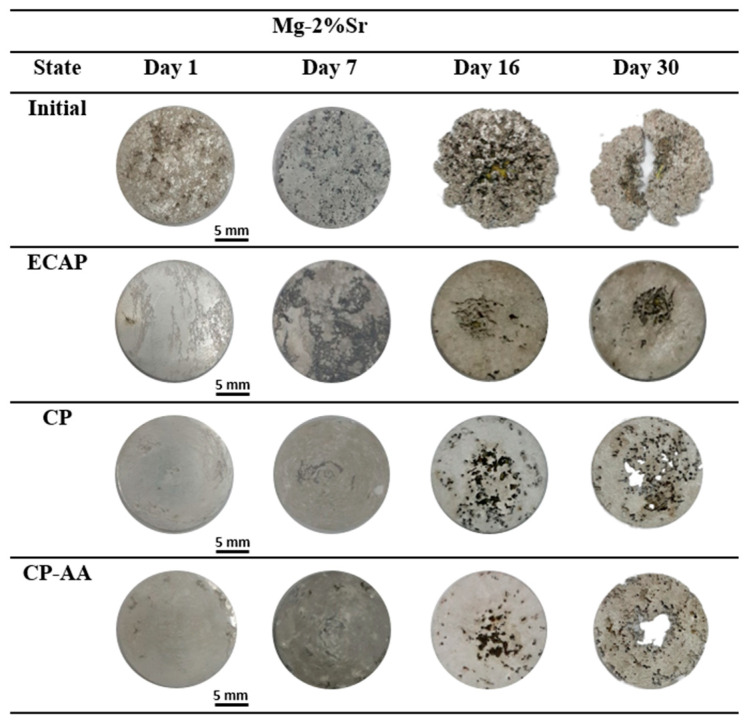
Images of specimens of the Mg-2Sr alloy in different structural states after immersion in Ringer’s solution over exposure time ranging from 1 to 30 days.

**Figure 7 materials-16-02279-f007:**
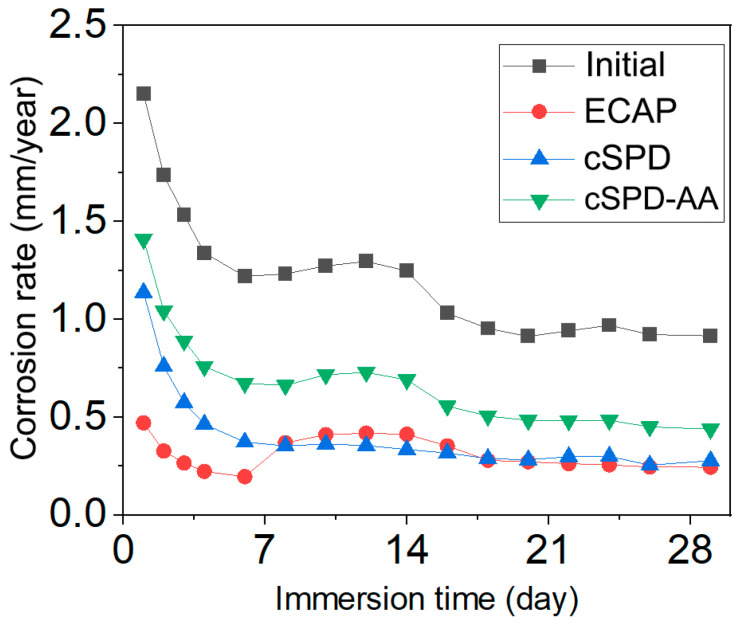
Corrosion rate of the Mg-2Sr alloy in different structural states after immersion in Ringer’s solution for exposure time up to 30 days.

**Figure 8 materials-16-02279-f008:**
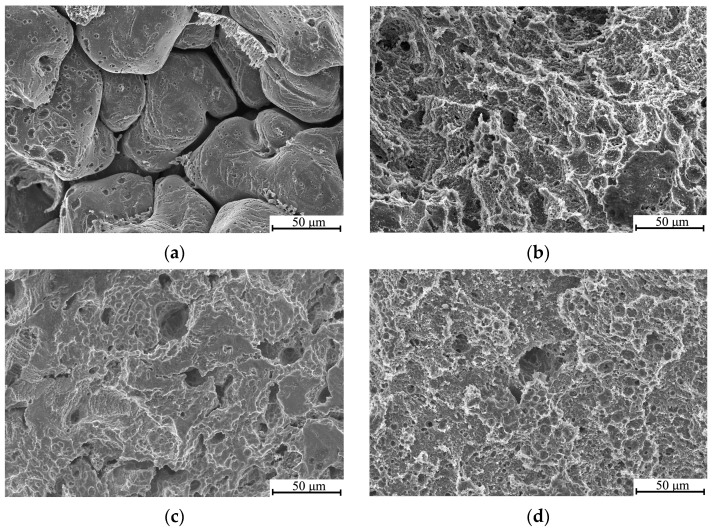
SEM images (in secondary electrons) of the surface of the Mg-2Sr alloy after an immersion time of 30 days: in the initial (**a**); ECAP (**b**); cSPD (**c**) and cSPD-AA (**d**) states.

**Table 1 materials-16-02279-t001:** The microstructure parameters calculated through the XRD analysis: the lattice parameters *a* and *c*, coherent domain size *D*, density of *a*-dislocations *ρ*, microstrain <*ε*^2^>^1/2^ and grain size *d* calculated from TEM images for the Mg-2Sr in different structural states.

State	Lattice Parameters, Å	*<ε*^2^*>*^1/2^, %	*ρ_a_*, m^−2^	*d_xrd_*, nm	*D*, μm
Initial	*a* = 3.2112 ± 0.0008 *c* = 5.2115 ± 0.00007	-	-	-	227 ± 21
ECAP	*a* = 3.2105 ± 0.0005 *c* = 5.2121 ± 0.0003	0.0385 ± 0.0005	1.7 × 10^15^	100 ± 15	10 ± 2
CP	*a* = 3.2081 ± 0.0003 *c* = 5.2090 ± 0.0001	0.0438 ± 0.0004	2.4 × 10^15^	76.1 ± 5	0.25 ± 0.05
CP-AA	*a* = 3.2098 ± 0.0002 *c* = 5.2115 ± 0.0003	0.0016 ± 0.0002	0.2 × 10^15^	245.8 ± 13	6.7 ± 1.7

**Table 2 materials-16-02279-t002:** Mechanical properties of the Mg-2%Sr alloy after different treatments.

State of the Mg-2%Sr Alloy	Microhardness, HV	YS, MPa	UTS, MPa	Elongation, %
Initial alloy	38 ± 5	63 ± 3	65 ± 4	0.7 ± 0.3
ECAP	55 ± 5	151 ± 15	193 ± 9	4.3 ± 0.5
CP	65 ± 6	225 ± 24	253 ± 25	0.2 ± 0.1
CP-AA	60 ± 6	180 ± 18	250 ± 23	6.5 ± 0.5

**Table 3 materials-16-02279-t003:** Corrosion rates of the Mg-2Sr alloy in different structural states.

State	Corrosion Rate, mm/y			
	Day 1	Day 3	Day 7	Day 30
Initial	2.15 ± 0.21	1.53 ± 0.12	1.27 ± 0.10	0.91 ± 0.13
ECAP	0.47 ± 0.04	0.26 ± 0.03	0.36 ± 0.02	0.24 ± 0.02
cSPD	1.14 ± 0.05	0.57 ± 0.02	0.35 ± 0.05	0.27 ± 0.05
cSPD-AA	1.41 ± 0.10	0.89 ± 0.08	0.66 ± 0.10	0.44 ± 0.09

## Data Availability

Not applicable.

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
