# Peer review of "Microstructural Assessment, Mechanical and Corrosion Properties of a Mg-Sr Alloy Processed by Combined Severe Plastic Deformation"

_materials, 2023, doi:10.3390/ma16062279_

Round 1

Reviewer 1 Report

I attached the review comment.

Reviewer 2 Report

The following are the comments

1.      Title of paper needs to be changed such as like this “Microstructural Assessment, Mechanical and Corrosion Properties of Mg-Sr alloy”.

2.      Have consistency of nomenclature such as if authors are writing Mg-Sr then it should go on like this in entire manuscript unless detailed once during synthesis.

3.      In general, English Language is poor; it must be improved in all aspects and respects.

4.      Introduction lacks the problem statement, I mean why this alloy was developed? I guess it is something more effective available in market.

5.      In general, SEM/TEM micrographs shows the charging effect which suggests that sample preparation is poor, therefore these images where charging or astigmatism is high needs to be replaced.

6.      What is the heat treatment cycle of the alloy produced?

7.      Show the tensile specimens instead of sample dimensions alone.

8.       

9.      Figure 6 at about 76 degrees, what is that compound except initial one? I mean what is that peak about?

10.  Figure 7, put the error bars and then discuss the deviation etc.

11.  For bone applications, the parameter of elastic modulus needs to be discussed and measured, please include it.

12.  Corrosion is the surface sensitive phenomenon, what is the surface roughness? OR how surface considerations were mitigated in manuscript which I do not find in manuscript. In addition, how surface area of the corrosion coupons was calculated?

13.  Conclusion seems to be huge which must be crispy and explicit of findings alone. I guess for this work one or two paragraphs would be sufficient.

Reviewer 3 Report

The subject matter of this work is interesting as well as the processing approach. However, it is required to the authors a significant revision of the contents before considering it for publication.

1. Please, consider reviewing the title that is a bit confused in the order. Something like "Study of the microstructure, mechanical and corrosion behavior of a Mg-Sr alloy after ECAP and HPT combined processing".

2. The term "temporary" has not to be associated with the term "bioresorbable". It only means that the function of the implant to the healing process has a fixed time. Once concluded, it can be removed because it represents a foreign device within the body with all the drawbacks you mentioned in the introduction. Many implants (screws, plates, pins) made of biostable metals (SS, Ti, CoCr,etc.) can be considered temporary in their function.

3. The experimental session must be improved. There are a lot of missing information. For instance, the atmosphere used during the homogenization and final annealing processes, the samples size before SPDs, the overall time of the corrosion tests, to cite some. Another important missing part is how the material was prepared, melting technique and procedure. 

4. Results session. It is not clear if SEM micrographs are collected by using SE (secondary electrons) or BSE (backscattering). Looking at the pictures it seems that some of them are by SE and some others by BSE. Please clarify.

5. Table 1. Why is reported Mg2Ca and Mg-1Ca. The work is focused on Mg-Sr alloy.

6. Please, explain in a clearer way why you relate the lattice parameters increase after CP-AA to a precipitation of the second phase. You do not have any further precipitation in this process, you simply observed a stress release (dislocations annihilation) and a little start of recrystallization (maybe). Moreover, with this statement you are anticipating the discussion in session 4. 

7. In paragraph 3.3 what does it mean "with increasing strain"? Please advice.

8. Corrosion is belonging to paragraph 3.4 and not 3.3. Please correct. Moreover, in this paragraph the corrosion test for CP sample is not described.

9. There also is a confusion in the microstructural description of SEM samples not related to the Fig. 10, but to Fig. 11. later in the paragraph you recover this error, but at the beginning it generates large confusion.

10. Please, improve the description of the graph reported in Fig. 10.

11. There is much more relevant literature on Mg-based alloys. Please review it by adding publications (to cite one, Prof. Norbert Hort).

12. English language must be improved a lot. There are typing mistakes and also statements that can better re-written.

Round 2

Reviewer 1 Report

The author answered most of my comments. Papers can be accepted for publication.

Author Response

We are thankful to the Reviewer for accepting our manuscript after the revision.

Reviewer 2 Report

Overall good. However, conclusion needs to be revised since it is too long. Also, it must be aligned with abstract too. 

Author Response

We thank the Reviewer for this remark. In accordance with this comment, we have reduced the conclusion by more than 1.5 times (from 28 to 17 lines) to make it more concise and compatible with the abstract (see attached manuscript with the changes after the 2nd revision highlighted in yellow). We hope that after this revision our paper can be positively evaluated for publication in the Materials journal. 

Reviewer 3 Report

Now, the work in the reviewed form is acceptable for publication.

Thanks all authors for the effort.

Author Response

We highly appreciate that the Reviewer found our paper suitable for publication in the revised form.